# Analysis of contemporary value and influence of intangible cultural heritage based on online review mining

Qiaoyun Xu[1], Yan Xu[2], Chao Ma[3]*

1 Normal School, Jinhua University of Vocational Technology, Jinhua, Zhejiang, China, 2 School of Marxism, Shanghai University of Finance and Economics, Yangpu District, Shanghai, China, 3 College of Economics and Management, Zhejiang Normal University, Jinhua, Zhejiang Province, China

* machao456@hotmail.com

## Abstract

### Purpose

The development of new media has enabled intangible cultural heritage to be disseminated through online platforms and attracted the attention of many contemporary young people. Classification and discussion on the value of intangible cultural heritage is an important way to help the inheritance and dissemination.

### Design/Methodology/Approach

Real online reviews were collected based on the Bilibili website as the research data source. A text-based BiGRU-Attention model was conducted to achieve value recognition and classification, and keyword statistics and topic analysis were performed for topic discussion among comments.

### Findings

Using the BiGRU-Attention model to classify intangible cultural heritage's contemporary values has a performance with precision, recall, and F1 score of more than 77%, the category of CVP (Cultural Value Perception) has the best classification performance. Through the topic analysis of comments and keywords, the cultural value of intangible cultural heritage is its core connotation, social value is the main purpose, and economic value is the power source.

### Originality

A BiGRU-Attention model based on BERT word embedding is proposed to achieve a text contemporary value perception recognition method based on user-generated content.

**Data Availability Statement:** All json files are available from the github page: https://github.com/SmartPrivate/ich_data/blob/main/message_word_cut.json.

**Funding:** 1.the National Natural Science Foundation of China (No. 72104219): Study design, data collection and analysis 2.the National Natural Science Foundation of China (No. 62207016): data collection and analysis 3.the MOE Project of Humanities and Social Sciences (No. 21YJC870013): preparation of the manuscript 4. Major Humanities and Social Sciences Research Projects in Zhejiang Higher Education Institutions (No. 2023QN129): preparation of the manuscript, decision to publish.

**Competing interests:** The authors have declared that no competing interests exist.

# Introduction

Intangible cultural heritage (ICH) refers to various oral traditions, performing arts, social customs, festivals, knowledge and skills, and related cultural spaces created and inherited by human society in the long-term historical development process [1]. ICH is an important part of world civilization, which embodies the historical context, cultural genes, and national spirit of all nations in the world [2]. ICH is not only a historical witness but also a contemporary resource. It embodies human wisdom and creativity, meets people's spiritual and emotional needs, and promotes social harmony and progress. However, unlike tangible cultural heritage, ICH is more likely expressed as a carrier of culture, which is facing the danger of disappearing and being forgotten with the impact of modernization and globalization and needs more attention and protection. In these processes, in addition to the traditional form of physical record and storage, digitalization and artificial intelligence technology have also played an important role in various aspects such as inheritance, dissemination, preservation, and presentation [3]. Thus, knowledge and content related to ICH can be disseminated more widely through the Internet in various formats such as text, images, audio, video, and even virtual reality, and augmented reality. This draws more and more attention to the ICH, especially among young people who are more familiar with new technologies.

However, as a new force in society, young people have different views and attitudes towards ICH. Some young people think that ICH is outdated, backward, and useless, and does not conform to the development of modern society, and some even are superstition and feudal remnants, which should be eliminated and discarded [4]. They lack understanding and respect for ICH, are unwilling to participate in and inherit it, and do not support relevant protection measures. Meanwhile, some young people believe that ICH is the root and soul of national culture, the witness and inheritance of history, and the source and driving force of innovation [5]. It does not contradict the development of modern society and can even promote and integrate. They have a strong interest and love for ICH, are willing to learn and inherit, and actively participate in and support related protection measures [6], even as a responsibility [7]. In the age of social media, these disunities of opinions manifest themselves in all kinds of User-generated Content (UGC) such as online reviews, posts, blogs, and so on. These contents contain a great number of young people's views, ideas, discussions, suggestions, and emotions, which is their most direct and true perception of the contemporary value of ICH. Therefore, to discover the opinions and emotions hidden in these UGCs, performing text mining and analysis is an effective way, and it is also possible to understand young people's real thoughts on ICH and even history and traditional culture, and their understanding of ICH in contemporary times. It is very meaningful research work in terms of inheritance, protection, and dissemination, and from these aspects, we conducted three research questions as follows:

1. What is the current state and distribution of users who publish topics and replies on social platforms?

2. Based on these topics and replies, how can we discover the posters' opinions and attitudes towards intangible cultural heritage?

3. Among these topics and reviews in different value aspects, what are the posters talking about online?

To address these issues, we will conduct research based on real Internet UGC. For the data we collected, first, we conducted a basic descriptive statistical analysis to reveal the distribution of the data set, and then, we built a usefulness recognition model for online reviews based on natural language processing methods to discover which reviews received the most attention

and were considered the most valuable. Finally, we used the topic analysis method based on keyword clustering to classify, display, and discuss these topics and content, and gave conclusions and suggestions.

## Related studies

### Contemporary value and influence of ICH

ICH is a resource for sustainable development. It can provide people with life skills, knowledge, and innovation capabilities, and can also bring benefits and opportunities for social and economic development [8]. Research on the contemporary value of intangible cultural heritage mainly covers three aspects: cultural value, social value, and economic value [9]. Cultural value also can be described as historical value, which reflects the deepest roots of a nation or group's traditional culture, retaining their life, way of existence, and the original state that formed their identity [10,11]. They show the way of thinking, psychological structure, aesthetic concepts, and traces of the cultural development of a nation or group [12]. Yan and Chiou [13] declared ICH is the symbol of the existence and the foundation of the development of a nation or group, and it is an important resource to continue the historical context, strengthen cultural self-confidence, and promote exchanges and mutual learning among civilizations. The social value of ICH has always been conducted in its relationship with social members and social parts [14]. Su, Li [15] thought ICH is the bond of social harmony, and it inherits the harmonious relationship between humans and nature, humans and society, and humans themselves, and promotes social stability and development. The research on the economic value of ICH is based on its integration with tourism, considering it as a sustainable tourism resource [16], and mainly focuses on the decision-making support role of ICH projects in the recommendation of tourist destinations [17]. Hsu, Zhang [18] constructed a multi-level conceptual model of ICH tourism among formal rationality and substantive rationality and found out it is influenced not only by altruistic intentions but also by the pursuit of egoistic benefits. Han and Bae [19] proposed an interesting perspective on how authenticity and nostalgia influence tourists' behavior in sharing ICH experiences on social media. Heredia-Carroza, Palma and Aguado [20] discussed the challenges of protecting ICH through copyright and examined how these regulations affect the perception and value of ICH, providing valuable legal context for the analysis of ICH in digital platforms. In addition to tourism, artistic creation based on ICH also has economic value. Heredia-Carroza, Palma Martos and Aguado [8] proposed an economic approach to measuring the value of flamenco performers, providing a quantitative perspective, and they also analyzed consumption habits of traditional music, providing relevant insights into the economic factors that influence the acquisition and perceived value of ICH [21].

### Opinion mining based on social media

Online comments have become an important way for people to express their opinions and share their experiences [22,23]. As the number of reviews increases dramatically, it becomes critical to classify and understand them effectively [24]. Opinion mining refers to the process of extracting information such as opinions, emotions, attitudes, and evaluations of authors or other entities from texts [25]. Opinion mining is an important research direction in the field of natural language processing, which has a wide range of application areas, such as public opinion analysis [26,27], product reviews [28–30] and social media [31–33].

Opinion mining based on social media faces some challenges, such as the unstructured, informal, and diverse content of social media [34], and the subjectivity, obscurity, and diversity of user opinions [35]. To address these challenges, researchers have proposed various methods, such as rule(aspect)-based methods [36,37], machine learning-based methods [38], and deep

learning-based methods [39,40]. With the development and improvement of language model research, such as Transformer and BERT, researchers can conduct more effective research based on more semantic information [41].

## Methodology

### ICH value perception classification framework

The framework of the whole research process is a typical pipeline of natural language text classification problems combined with subcategory topic analysis and discussion which is shown in Fig 1.

We designed the dimensions of the classification based on Hofstede's theory of cultural value [42]. Hofstede showed the value of culture in his research, especially in various forms of communication and exchange. In his latest research, he divided the value of culture into six dimensions, which are related to society, culture and economy [43]. Specifically, we also combined the characteristics of ICH and summarized the literature from cultural, legal, and economic perspectives and frameworks [13,14,20]. Therefore, based on his theory of cultural value, we divided the value perception in online reviews into four categories: *cultural value*

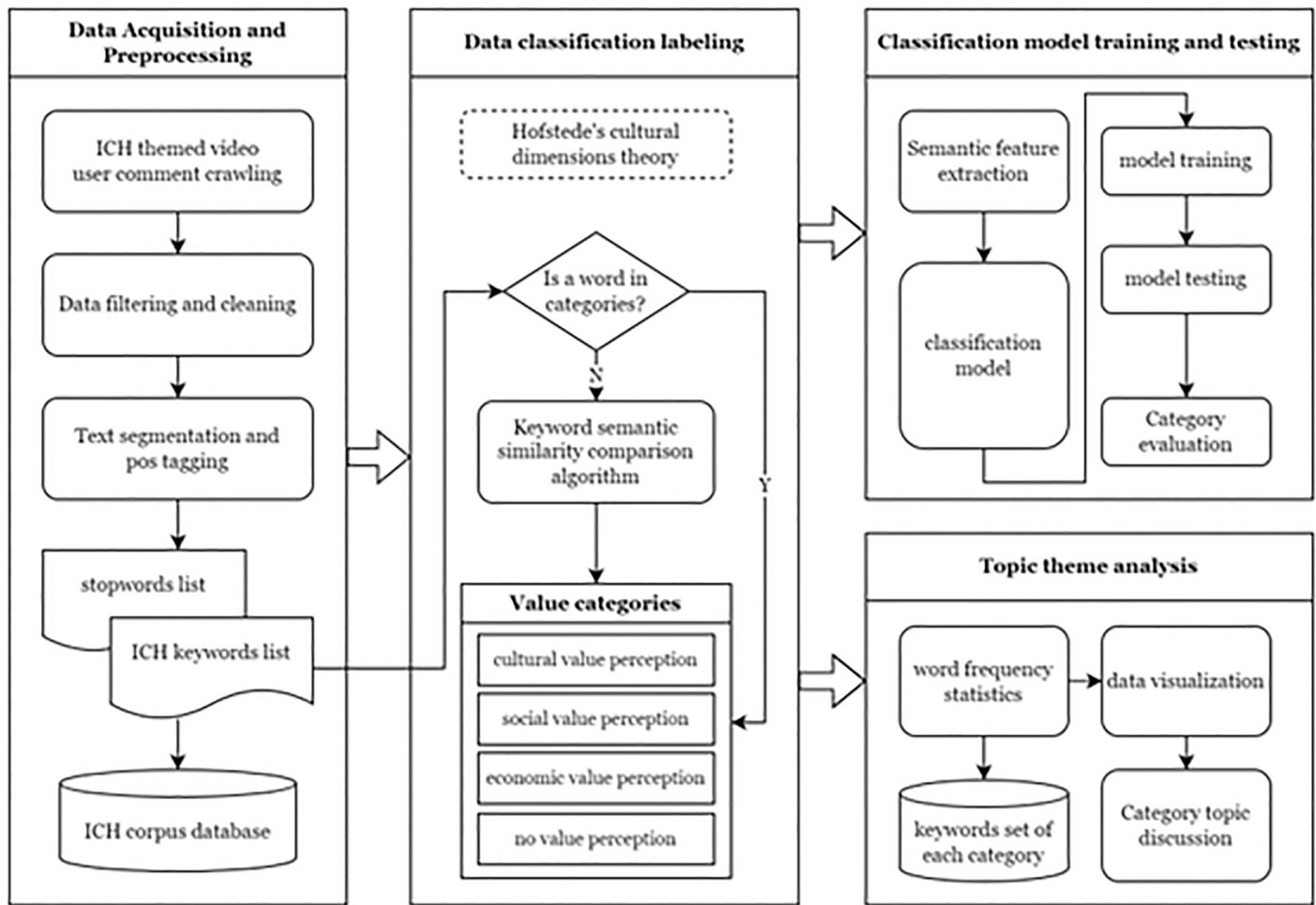

**Fig 1. ICH value perception recognition and classification pipeline.**

*perception*, *social value perception*, *economic value perception* and a category that does not include value perception, labelled *no value perception*.

The first step is the cleaning of the original dataset, the expressions and symbols without clear meaning are filtered, and the text content with better quality is retained. Secondly, due to the coherence of Chinese text, to achieve word-level embedding, a word segment process is necessary. Meanwhile, to ensure the accuracy of word segmentation results and improve the semantic quality of words, we imported Chinese stop word lists to remove stop words during the word segmentation process and used a pre-built intangible cultural heritage keyword list as a user dictionary. We used the BERT model to embed these keywords [44]. In the part of model training and model testing, we chose the supervised learning method based on deep learning and machine learning for the text multi-category classification tasks that this research conducted. The data set is divided into a training set and a test set by random sampling, and then model parameter training is carried out. For the result, we use the macro average to evaluate the accuracy, recall, and F1 score of the model, and the classification results for each category. Finally, based on the classification results, we conducted a detailed topical discussion. Based on the content contained in each category, we conducted a statistical analysis of word frequency, using high-frequency words as keywords, and further explored the perception of contemporary ICH value around these keywords.

### Research data

**Data collection.**   To ensure the reliability of our research, we set two basic rules for the selection of data sources. The first one is authenticity, we must choose real UGC from the Internet; The second one is rejuvenation, the data should come from a website with young people or the younger generation as the main user group. Of course, this data should be easily accessible. We chose to obtain data from Bilibili with a Python-based web crawler. The core content of Bilibili is self-media sharing, which contains many high-quality user-made video content with rich online comments and valuable social behaviors. For the video itself, you can operate "like", "coin", "favorite" and "comment" and for each comment, you can operate "like" and "reply". These characteristics are very suitable for online review mining.

We use " intangible cultural heritage " as the keyword to search the website by topic, excluding articles, documentaries, music, and other types, and only keep user-made videos as search results. We ended up with a record of 978 videos, and these records contained a total of 127,470 comments. However, not all these reviews are worthwhile. Benefits from the comment usefulness evaluation system provided by Bilibili, comments that are considered valuable by everyone will be marked by a quantitative indicator "like". Therefore, we exclude records that get no "like" mark and only one "like" mark which is usually marked by the poster himself. Finally, 23,130 records for classification and topic analysis were obtained.

**Data labelling.**   To discover the reviewer's understanding of the value conveyed by the video content from online reviews, we conducted a supervised learning text classification model. According to previous research, for the contemporary value perception of ICH, we annotated the data into four categories: "cultural value perception", "social value perception", "economic value perception" and "no value perception". We used unsupervised automatic annotations combined with manual labeling to counter our large-scale data set. The unsupervised automatic labeling process contains three steps. Firstly, we built three-word tables, each containing a series of keywords in three categories. For example, the "cultural value perception" word table contains words like "culture, cultural, traditional, historical" and the "social value perception" contains words like "activity, society, meeting, greeting".

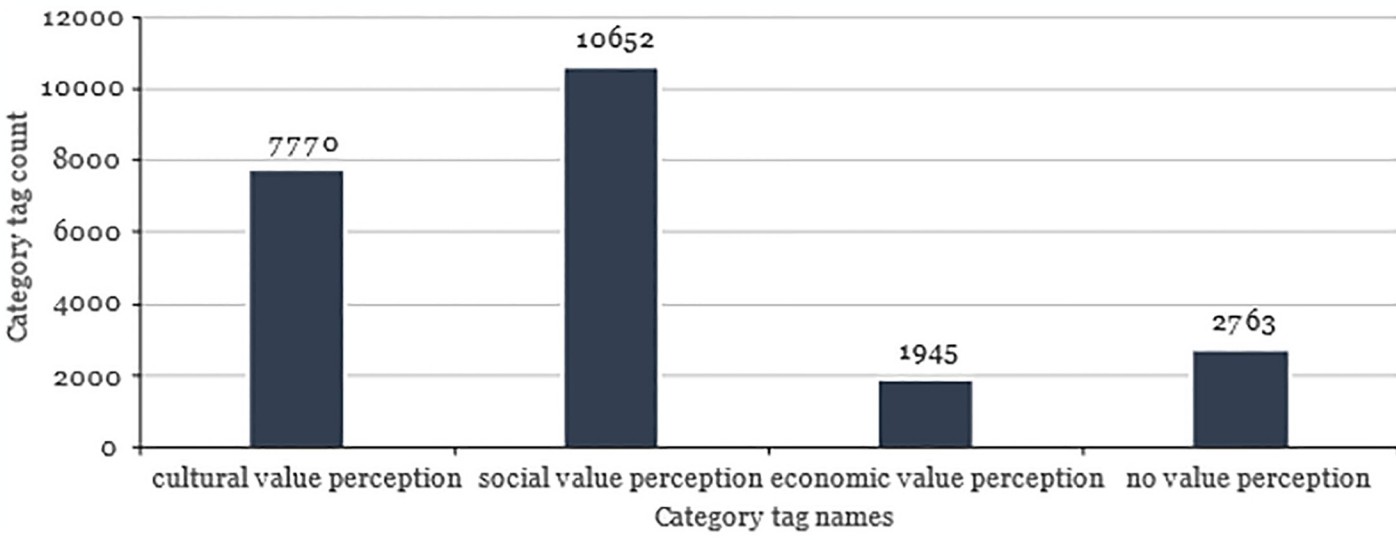

**Fig 2. Labeling results of ICH value perception classification.**

Second, we segment the original online review text into several words and match them with the words in the vocabulary. If the text contains keywords of the specific category, this record would be labeled as the current category. Of course, we can't build a vocabulary that includes all keywords. Therefore, for those records that cannot be matched to the category, we use a text similarity comparison algorithm of the word semantic space vector to calculate the category with the highest average semantic cosine similarity with the keywords in the three categories as the basis for classification.

The third stage is the final inspection of the classification results, which is processed manually. The records' classification results containing multiple keywords should be annotated manually by three experts in the field of ICH. Finally, the data annotation results of four categories are shown in Fig 2.

## BBGA (Bert-BiGRU-Attention) classification model

To effectively identify the topic tendencies contained in the text, it is necessary to build a classification model based on natural language text sequences based on semantic features. We propose a BBGA model to achieve this goal, which is shown in Fig 3. The BERT model is used to implement word vector embedding of text. The BERT model refers to the Bidirectional Encoder Representations from Transformers, which provides richer and more accurate word embeddings by considering contextual information and utilizing bidirectional encoding. Compared to Word2Vec, the advantage of BERT in word embedding lies in its ability to consider richer contextual information, achieving more accurate semantic representations, rather than relying solely on local window-based co-occurrence statistics [45]. The Bi-GRU model [46] of a bidirectional recurrent neural network containing a threshold recursive unit is used to extract sequential features and temporal features in the text. And combined with the attention mechanism [47] to process the final feature weights. Finally, the softmax classifier is used to achieve multi-classification based on the data labelling results.

**Bi-GRU (Bidirectional Gated Recurrent Unit).** The task of identifying and classifying contemporary ICH value perception from review content is essentially a text sequence

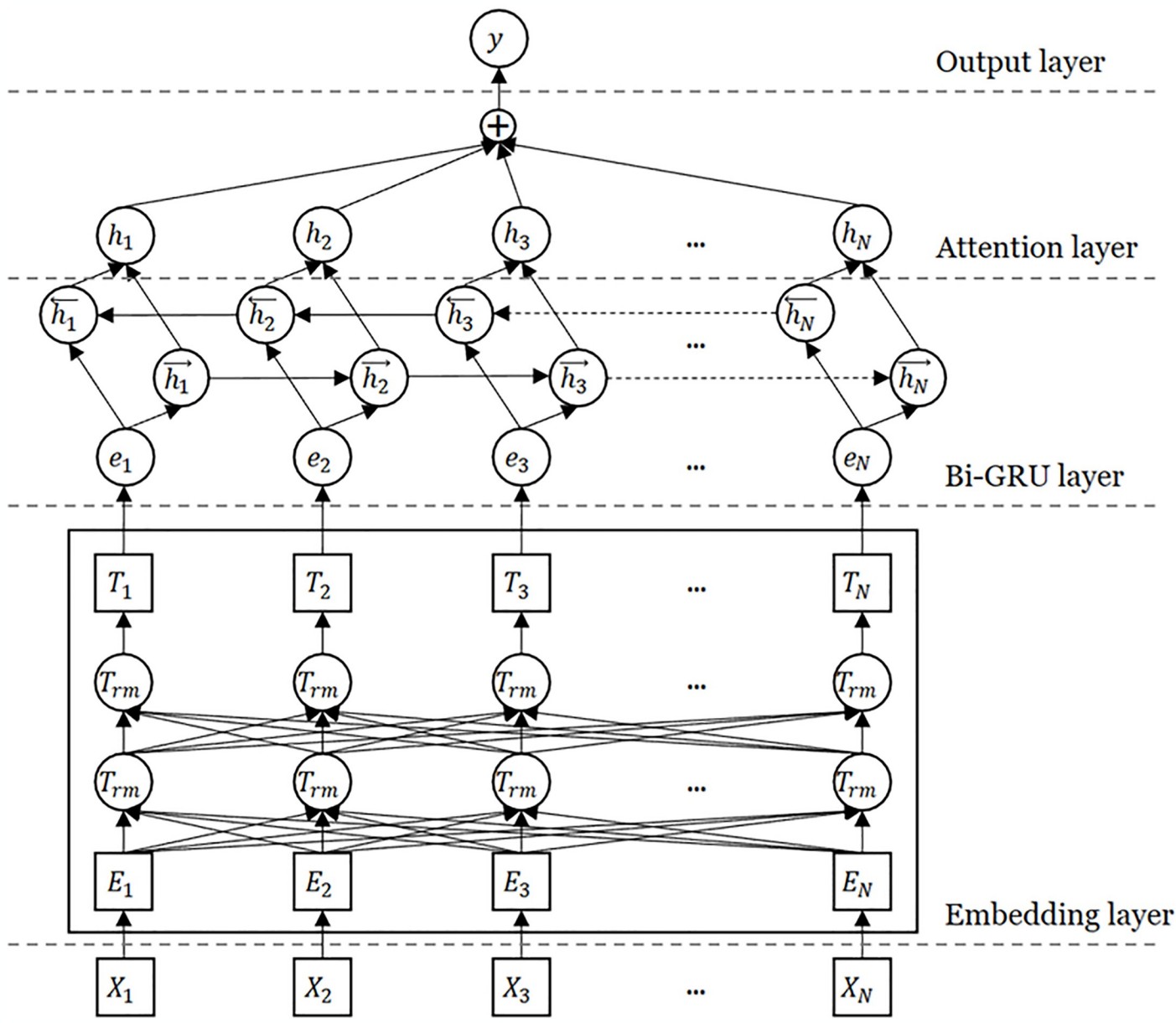

**Fig 3. Main structure of BBGA model.**

classification problem based on the semantic features of natural language corpus. Among numerous machine learning and deep learning models and frameworks, the Recurrent Neural Network (RNN) architecture is considered the most suitable for this type of task. LSTM (Long-short Term Memory) and GRU (Gated Recurrent Units) are both optimized recurrent neural network models [48], suitable for processing long sequences in sequence models. Compared with the traditional RNN model, an update mechanism is added, the memory ability is enhanced, and the chances of gradient disappearance and gradient explosion are reduced. Both models have large-scale applications in text classification, machine translation, and speech recognition. As an improvement of the LSTM model, the GRU model simplifies the

input gate and forget gate into an update gate in units of hidden layers [49]. A gated recurrent unit contains an update gate $z_t$ and a reset gate $r_t$ which can be present as:

$$z_t = \sigma(W_z \times [h_{t-1}, x_t])$$

$$r_t = \sigma(W_r \times [h_{t-1}, x_t])$$

$t$ means current state time. At time $t$, $h_{t-1}$ means the output of previous state, $x_t$ means current input. W means the weight matrix of the model. After the gated state of the previous stage is input to the unit, the reset gate can intervene to handle this state, which can be calculated as an updated ($h_{t-1} \times x_t$), then append this status to the new data and combine it with the current input $x_t$. Finally, the new state $h_t$ of a gated recurrent unit can be calculated as:

$$h_t = tanh(W_r \times [h_{t-1} \times r_t, x_t])$$

$tanh$ is the activation function. The present hidden state $h_t$ is derived from the current input $x_t$, which also acts as the prospective hidden layer. The variable $r_t$ can modulate the extent of memory utilized, amalgamating the current hidden state $h_t$ with the preceding hidden state $h_{t-1}$ to deduce the ultimate state of the hidden layer. The computation for the current hidden state h_t is hence revised as follows:

$$h_t = (1 - z_t) \times h_{t-1} + z_t \times h_t$$

$z_t$ means the gate control signal which controls the forgetting volume of information and locates in $z_t \in [0,1]$, to measure the extent to which information needs to be forgotten.

## Result discussion

### Descriptive statistical analysis

This dataset consists of 803 observations. The mean, or average, value is 787.43. The standard deviation, which measures the variability or spread of the data, is 6320.90. The minimum value in the dataset is 1, while the maximum value is 169728.

The 25th percentile, also known as the first quartile, is 9. This means that 25% of the observations in the dataset are less than or equal to 9. The 50th percentile, or the median, is 85. This implies that 50% of the observations fall below or equal to 85. The 75th percentile, or the third quartile, is 402. Thus, 75% of the observations are less than or equal to 402.

### Discussion of value perception classification results

We used the commonly used classification model confusion matrix evaluation method to evaluate the classification results and counted the accuracy, recall, and F1 score. To ensure the robustness of the classification model, we also selected some other SOTA classification models for comparison. The BERT model is a pre-trained language model, so it also can complete text classification tasks. Text-RCNN is a composite model that combines CNN and RNN in text classification tasks and has good classification capabilities [50]. Besides, we used a common GRU model as the baseline deep learning model, Multinomial Naive Bayes, and support vector machine (SVM) as the baseline machine learning model. The results of the comparative experiments are presented in Table 1. Compared with several commonly used machine learning and deep learning classification models, the BBGA model we built has the best score in macro average, which can reach 0.79 in precision and 0.78 in F1 score. The BERT model has almost the same performance as the BBGA model, benefiting from its powerful pre-training scale. The contextual features are very important for text classification.

**Table 1. Performance comparison for each model.**

| Model | Indicators | Overall (Macro average) | Categories | | | |
|---|---|---|---|---|---|---|
| | | | CVP | SVP | EVP | NVP |
| BBGA | Precision | 0.79 | 0.85 | 0.82 | 0.73 | 0.76 |
| | Recall | 0.77 | 0.86 | 0.84 | 0.68 | 0.68 |
| | F1 | 0.78 | 0.85 | 0.83 | 0.70 | 0.72 |
| BERT | Precision | 0.79 | 0.84 | 0.83 | 0.75 | 0.74 |
| | Recall | 0.77 | 0.86 | 0.78 | 0.74 | 0.68 |
| | F1 | 0.77 | 0.85 | 0.85 | 0.73 | 0.63 |
| Text-RCNN | Precision | 0.75 | 0.80 | 0.71 | 0.75 | 0.73 |
| | Recall | 0.73 | 0.73 | 0.73 | 0.74 | 0.71 |
| | F1 | 0.72 | 0.74 | 0.70 | 0.75 | 0.68 |
| GRU | Precision | 0.75 | 0.76 | 0.74 | 0.76 | 0.73 |
| | Recall | 0.73 | 0.74 | 0.72 | 0.74 | 0.71 |
| | F1 | 0.72 | 0.73 | 0.71 | 0.73 | 0.70 |
| Multinomial NB | Precision | 0.65 | 0.66 | 0.64 | 0.66 | 0.63 |
| | Recall | 0.64 | 0.65 | 0.63 | 0.65 | 0.62 |
| | F1 | 0.62 | 0.63 | 0.61 | 0.63 | 0.60 |
| SVM | Precision | 0.72 | 0.73 | 0.71 | 0.73 | 0.70 |
| | Recall | 0.70 | 0.71 | 0.69 | 0.71 | 0.68 |
| | F1 | 0.71 | 0.72 | 0.70 | 0.72 | 0.69 |

The Text-RCNN model needs to rely on text for training, so even if the model design is good enough, it still reflects the gap with the classification model based on the pretrained language model. The GRU neural network model with Tencent Chinese word2vec embedding can also score good results, which is the advantage of the recurrent neural network in processing text classification tasks including context. However, due to the completion and truncation of the variable length of the corpus, some features are lost, which may have an impact on the results. On the other hand, the classification effect of the two machine learning methods is not very significant. Naive Bayes and support vector machine algorithms may be more suitable for binary classification problems with low-dimensional data. Meanwhile, comparing the classification results of each category, among the three categories of Cultural Value Perception (CVP), Social Value Perception (SVP), and Economic Value Perception (EVP), the classification effect of CVP performed best.

We discuss the classification results of each category in more detail, especially the results of classification errors. We found that in the classification results of cultural value, the main meaning of the wrong records was "worth learning or worth experiencing". Sentences that contained the semantic meaning of "worth" and expressed relatively short sentences were wrongly classified into the economic value category. Meanwhile, in social value classification, a common classification error is recorded as "helping spread or increasing communication", keywords like "help" or "increase" can lead to incorrect classification as cultural value or economic value. In addition, we usually believe that the classification of economic value should be the most accurate, because the economic value of intangible cultural heritage is reflected in cultural creation and tourism, and its key expressions usually include "buy or play", which has obvious verb characteristics. However, the classification results in this study performed the worst. After detailed analysis, we found that these expressions that reflect economic value usually contain and combine expressions of cultural value and social value, so the features are hidden, which confused the classifier.

## Discussion of comments' topic analysis results

By using word frequency statistical analysis methods, we conducted thematic analysis from the perspective of contemporary value perceptions of three categories of ICH. Among these categories, the comments posted by netizens have different emphases, but they also have similarities. Discussions on the cultural value of ICH mainly focus on its core connotation. Everyone in the comments generally believes that ICH is of great significance to the inheritance of history and culture. Meanwhile, there are many discussions about the social value of ICH, the existence of ICH is more like a bond between different types of social members forming common cognition and value recognition. Finally, the value of ICH is also reflected in its economic aspects. From cultural and creative products to themed tourism, it has gradually become an activated embodiment of traditional culture, which is becoming a fashion, and has great market potential. Based on the classification results, we carried out specific discussions on the value expression of these three aspects.

**Cultural value of ICH: Core connotation.** The word cloud generated by topics classified as Cultural Value Perception is shown in Fig 4, and high frequency words are shown in Table 2. We can find that comments containing the keyword "art" appear the most times since art is an important form of presentation of culture. In addition, expressions such as "opera" and "drama", as well as the materialization of traditional skills such as "clothing", are also frequently mentioned.

ICH is an important part of human cultural diversity. It reflects the characteristics and features of different nations, regions, and communities, and demonstrates the richness and diversity of human culture. ICH is also an important carrier of cultural inheritance and innovation. It carries people's cognition and understanding of life, nature, and society, conveys people's pursuit and yearning for beauty, kindness, and justice, and stimulates people's interest in

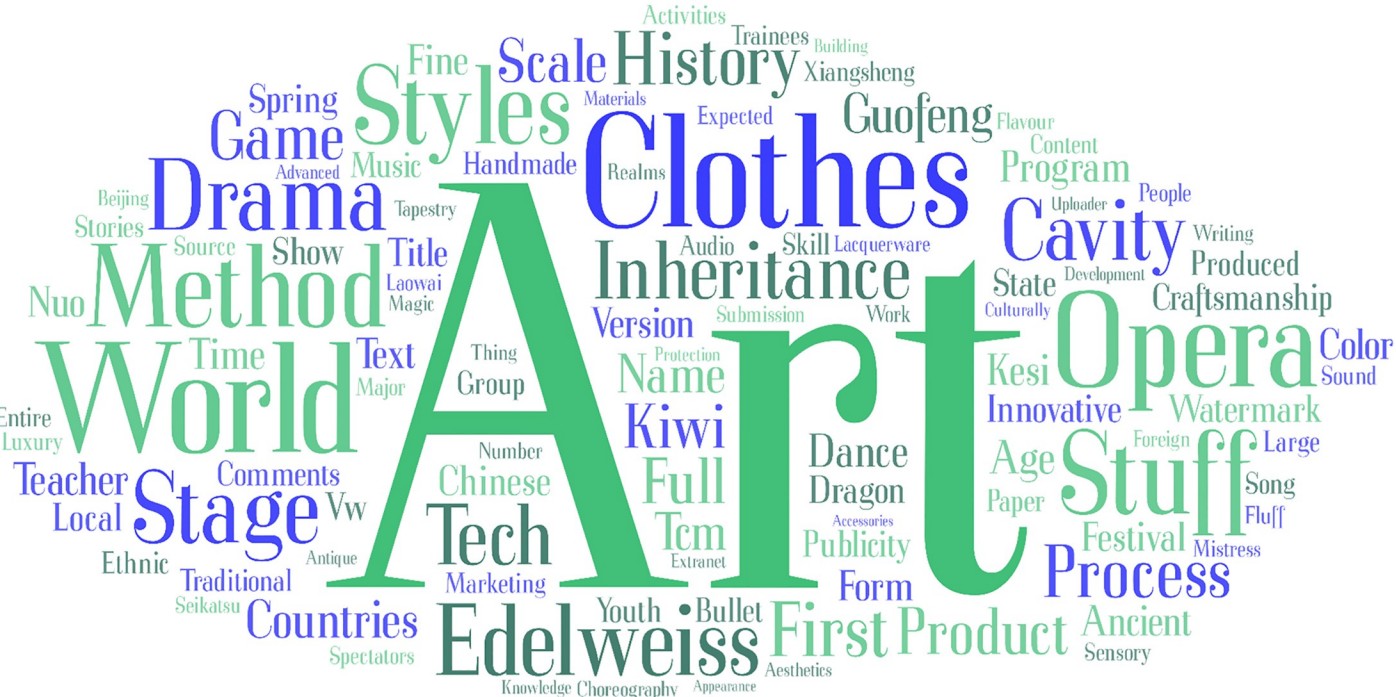

**Fig 4. Word cloud of cultural value of ICH.**

**Table 2. High frequency keywords classified as cultural value perception (Top 20).**

| Word | Frequency | Word | Frequency |
|---|---|---|---|
| Art | 654 | Drama | 140 |
| Clothes | 366 | Cavity | 139 |
| World | 335 | History | 135 |
| Method | 329 | Process | 131 |
| Opera | 305 | Tech | 124 |
| Styles | 248 | Product | 114 |
| Edelweiss | 238 | Guofeng | 112 |
| Stuff | 214 | Countries | 111 |
| Stage | 210 | Craftsmanship | 108 |
| Inheritance | 210 | Game | 108 |

cultural traditions. Respect and inheritance, as well as the exploration and practice of cultural innovation. On the other hand, ICH is the product of a specific cultural and geographical environment and the background of the times [51]. It depicts the production, lifestyle, and cultural traditions of a country or a place. It is an identification mark for the establishment of cultural identities in various regions. It has rich and colorful cultural forms and profound cultural connotations, including historical value, symbolic value, spiritual value, and aesthetic value [52]. These are important cultural resources in contemporary society and play an important role in protecting cultural diversity. Meanwhile, ICH is also a variety of traditional cultural expressions that exist in non-material forms that are closely related to the production and life of the masses and are loved by the masses. They can enrich the daily cultural life of the people, broaden people's horizons, cultivate their sentiments, increase their traditional cultural knowledge, and satisfy their spiritual needs [2]. Therefore, cultural value is the core connotation of ICH, and it is also the most attractive part.

**Social value of ICH: Main purpose.** The word cloud generated by topics classified as Social Value Perception is shown in Fig 5, and high frequency words are shown in Table 3. In comments and discussions about social value, the most frequently mentioned keywords are "world", and so did "countries", "people", and "popular", which are all about the role of association and binding among human beings worldwide.

Discussion topics about the social value of ICH mainly focus on the social role. On one hand, as an expression of living culture, contemporary young people are willing to use it as a common topic through the perception of the form of intangible inheritance, transmission mechanism, and spiritual connotation, forming groups or groups, organizing activities, and enhancing communication [53]. On the other hand, there are also many discussions about ICH social activities, mainly focusing on the means and ways how ICH enters the community and socialization experience [54,55]. As we all know, ICH is an important source of social cohesion and identity. It enhances people's sense of belonging and pride in their own nation, region, and community, promotes exchanges and interactions between different groups, and enhances the relationship between people [4]. Understanding and respect maintain social stability and unity. ICH is also an important means of social education and guidance [56]. It spreads people's recognition and compliance with morality, law, and public order, cultivates people's responsibility and participation in social responsibility and civic awareness, and shapes people's orientation towards social development.

**Economic value of ICH: Power source.** The word cloud generated by topics classified as Economic Value Perception is shown in Fig 6, and high frequency words are shown in

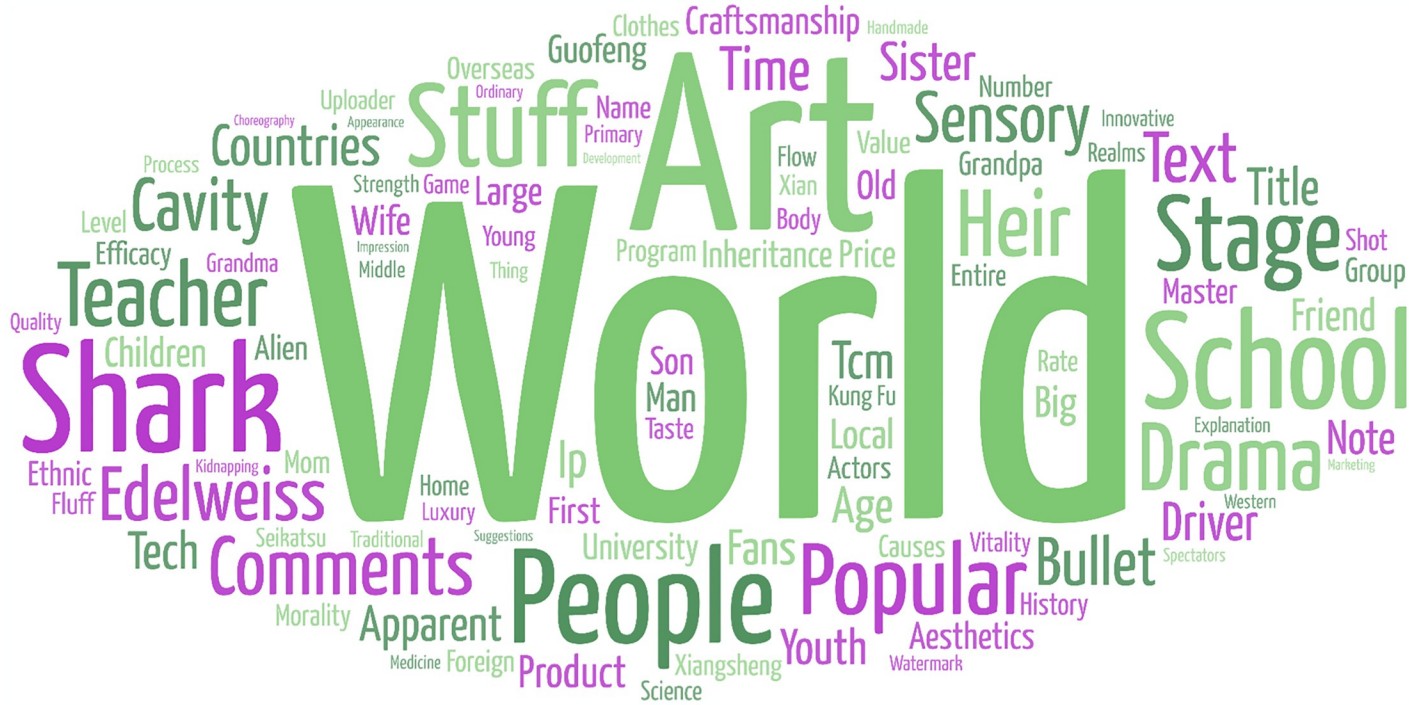

**Fig 5. Word cloud of social value of ICH.**

Table 4. Among the comments classified as economic value, the most frequent word is "product". The materialization and productization of ICH are what young people are most concerned about. In addition, words related to marketization such as "cost", "marketing" and "brand" were mentioned many times.

ICH is an important driving force for economic development and innovation. It provides rich knowledge and skills, product resources and market demand, provides new opportunities and space for all walks of life, and injects new vitality and impetus into economic growth [57]. ICH is also an important support for tourism and related industries [58]. It attracts many domestic and foreign tourists, drives the development of catering, accommodation, transportation, shopping, and other related industries, and brings huge benefits and benefits to the

**Table 3. High frequency keywords classified as social value perception (Top 20).**

| Word | Frequency | Word | Frequency |
|---|---|---|---|
| World | 317 | Sensory | 104 |
| Art | 259 | Edelweiss | 102 |
| People | 239 | Drama | 101 |
| Shark | 199 | Cavity | 101 |
| School | 163 | Countries | 101 |
| Popular | 149 | Heir | 100 |
| Stuff | 145 | Apparent | 100 |
| Comments | 144 | Craftsmanship | 94 |
| Teacher | 114 | Bullet | 91 |
| Stage | 107 | Text | 85 |

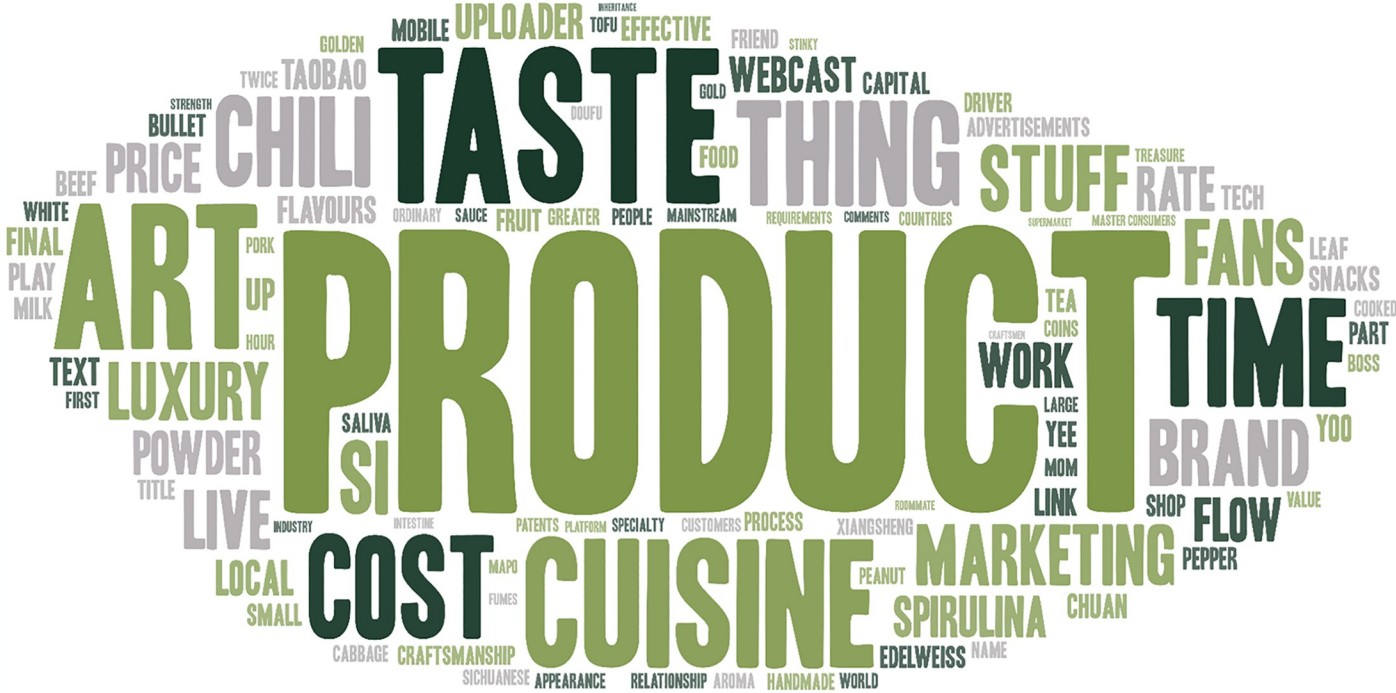

**Fig 6. Word cloud of economic value of ICH.**

local economy. Economic value is the value that can be measured by economic indicators, and it is an important driving force for ICH to enter the modern market and realize its activation. ICH is scarce, and all walks of life hope to enrich the modern product market and form a diversified and personalized product system with its market development. it is necessary to enhance the cultural taste of ICH products and to achieve both refined and popular [59]. Expand the market share of ICH products, to obtain more economic benefits and increase the income of ICH business, drive employment in the cultural and creative industry, and expand employment opportunities. The huge economic value contained in ICH products makes more and more enterprises willing to participate in the ranks of its industrialization, which is conducive to the increase of the income of ICH practitioners and is also conducive to dissemination

**Table 4. High frequency keywords classified as economic value perception (Top 20).**

| Word | Frequency | Word | Frequency |
|---|---|---|---|
| Product | 194 | Stuff | 84 |
| Taste | 166 | Luxury | 83 |
| Art | 50 | Fans | 82 |
| Cuisine | 50 | Spirulina | 79 |
| Thing | 45 | Powder | 77 |
| Marketing | 42 | Price | 77 |
| Cost | 38 | Uploader | 72 |
| Time | 33 | Live | 71 |
| Chili | 28 | Webcast | 70 |
| Brand | 27 | Work | 67 |

and development. How to realize its economic value through development and utilization since effective protection and inheritance of ICH, to make it a cultural value-added is an important challenge.

## Practical implications and recommendations

This study offers valuable insights for cultural policymakers, heritage managers, and academics. For policymakers, understanding how ICH is discussed online can inform policies that balance preservation with accessibility, ensuring ICH remains authentic while reaching global audiences. For heritage managers, the analysis highlights the importance of digital strategies in ICH preservation. By understanding public sentiment from online reviews, managers can create more targeted outreach and engagement efforts that resonate with diverse audiences. Furthermore, for academics, this research provides a new methodology for studying ICH in the digital age, offering a foundation for further interdisciplinary research on the relationship between culture, technology, and society. In summary, the findings can guide more informed policy development, enhance heritage management practices, and open new research avenues, supporting the sustainable preservation and promotion of ICH in the digital environment.

## Limitations and future research

While this study provided a broad analysis of self-media websites using intangible cultural heritage (ICH) as a keyword, the scope was relatively general, and more specific research is needed. Future studies could focus on particular types of ICH, using topic mining and opinion analysis to provide deeper insights into how specific forms of ICH are perceived online.

Additionally, online reviews are subject to various biases, such as demographic factors, emotional attachments, and platform algorithms, which can skew the representation of ICH. Future research should address these biases and explore how they might influence public perceptions of ICH. Advanced techniques such as sentiment analysis could be applied to better identify patterns in the data and mitigate these biases. By incorporating these methods, future studies could validate and extend the current findings, offering a more nuanced understanding of ICH in the digital environment.

## Conclusion

Under today's social reality, compared with previous historical stages, intangible cultural heritage has changed to a certain extent in terms of actual functions, social significance, and presentation methods [3]. With the development of science, the advancement of technology, and the evolution of history, the practical functions of some items may be weakened. But that doesn't mean it loses its value and significance to us today. Our research is based on Hofstede's theory of cultural value dimensions, combined with the current cultural cognition of young people, and divides the value of intangible cultural heritage to contemporary society into three categories: economic value, cultural value, and social value. Unlike physical value or monetary value, the value of intangible cultural heritage is mostly subjective. The significance lies in the maintenance of emotions and the inheritance of culture. This emotion is often reflected in the comments and expressions about intangible cultural heritage. Thanks to natural language processing technology and machine learning classification methods, we can analyze and explore these hidden emotions and values from a semantic level. For the different topic themes obtained, we can absorb the opinions and suggestions in them and play a role in the milestone inheritance of intangible cultural heritage, which is a treasure of human history.

In the historical period of globalization, urbanization, digitization, and rapid changes in science and technology, intangible cultural heritage especially highlights its necessity,

importance, and infinite charm. It mostly remains an important and central component of our way of life [60]. In addition, intangible cultural heritage has especially demonstrated its great practical significance in terms of deepening the sense of identity, enhancing the sense of history, and enhancing the sense of happiness.

First, intangible cultural heritage is an important basis for enhancing our national identity, the root of our culture and the soul of the nation, the embodiment of our national characteristics and national character, and it is also a sign for other nations to understand the Chinese nation. This sense of identity enables each of us to find our place in society, where our cultural space is, and where our relatives, friends and social groups live. Therefore, we have reliance, strength, joy in life and meaning of life [61].

Secondly, the overall space of intangible cultural heritage in which we live has a long history and glorious multiple imprints of the times. Through intangible cultural heritage, we connect ourselves with the cultural traditions created and passed down by generations of ancestors, and we become part of this cultural tradition [62]. To a certain extent, this cultural tradition regulates our behavior, how we deal with people, our emotions, and our value judgments. With this sense of history, we have enhanced our nation's self-confidence and sense of pride.

## Author Contributions

**Conceptualization:** Qiaoyun Xu.

**Data curation:** Chao Ma.

**Formal analysis:** Yan Xu.

**Methodology:** Chao Ma.

**Software:** Chao Ma.

**Visualization:** Chao Ma.

**Writing – original draft:** Qiaoyun Xu.

**Writing – review & editing:** Yan Xu.

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
