## [Decision Letter · Decision Letter 0]

22 Mar 2024

PONE-D-24-01376Analysis of Contemporary Value and Influence of Intangible Cultural Heritage Based on Online Review MiningPLOS ONE

Dear Dr. Ma,

Thank you for submitting your manuscript to PLOS ONE. After careful consideration, we feel that it has merit but does not fully meet PLOS ONE’s publication criteria as it currently stands. Therefore, we invite you to submit a revised version of the manuscript that addresses the points raised during the review process.

We look forward to receiving your revised manuscript.

Kind regards,

Wei-Yen Hsu, Ph.D.

Academic Editor

PLOS ONE

“1.the National Natural Science Foundation of China (No. 72104219)

2.the National Natural Science Foundation of China (No. 62207016)

the MOE Project of Humanities and Social Sciences (No. 21YJC870013)

3.Major Humanities and Social Sciences Research Projects in Zhejiang Higher Education Institutions (No. 2023QN129).”

4. In the online submission form, you indicated that your data will be submitted to a repository upon acceptance.  We strongly recommend all authors deposit their data before acceptance, as the process can be lengthy and hold up publication timelines. Please note that, though access restrictions are acceptable now, your entire minimal  dataset will need to be made freely accessible if your manuscript is accepted for publication. This policy applies to all data except where public deposition would breach compliance with the protocol approved by your research ethics board. If you are unable to adhere to our open data policy, please kindly revise your statement to explain your reasoning and we will seek the editor's input on an exemption.

5. We note that Figures 4 and 5 in your submission contain copyrighted images. All PLOS content is published under the Creative Commons Attribution License (CC BY 4.0), which means that the manuscript, images, and Supporting Information files will be freely available online, and any third party is permitted to access, download, copy, distribute, and use these materials in any way, even commercially, with proper attribution. For more information, see our copyright guidelines: http://journals.plos.org/plosone/s/licenses-and-copyright.

1. You may seek permission from the original copyright holder of Figures 4 and 5 to publish the content specifically under the CC BY 4.0 license.

Reviewers' comments:

Reviewer's Responses to Questions

**Comments to the Author**

1. Is the manuscript technically sound, and do the data support the conclusions?

Reviewer #1: Yes

Reviewer #2: Yes

Reviewer #3: Partly

2. Has the statistical analysis been performed appropriately and rigorously? 

Reviewer #1: Yes

Reviewer #2: Yes

Reviewer #3: No

3. Have the authors made all data underlying the findings in their manuscript fully available?

Reviewer #1: Yes

Reviewer #2: Yes

Reviewer #3: No

4. Is the manuscript presented in an intelligible fashion and written in standard English?

Reviewer #1: Yes

Reviewer #2: Yes

Reviewer #3: Yes

5. Review Comments to the Author

Reviewer #1: Accept the Article entitled "Analysis of Contemporary Value and Influence of Intangible Cultural Heritage Based on Online Review Mining". This article addressing Influence of Intangible Cultural Heritage Based on Online Review Mining.

Reviewer #2: *Valuable paper for this era.

*The examination of ICH's current significance and impact reveals

its varied function in fostering innovation, social cohesiveness,

cultural diversity, and sustainable development. Communities may

take advantage of ICH's potential to improve lives and build a more

inclusive and sustainable future by realising its significance and

supporting its promotion and preservation.

Reviewer #3: The authors have used machine learning techniques for ICH (Intangible Cultural Heritage) value perception classification task. The manuscript lacks substantial data and methodology, such as details on neural network architectures (e.g., BERT, Bi-GRU RNN).

6. PLOS authors have the option to publish the peer review history of their article (what does this mean?). If published, this will include your full peer review and any attached files.

Reviewer #1: **Yes: **BALAPRAKASH VADIVEL

Reviewer #2: No

Reviewer #3: No

---

## [Author Response · Author response to Decision Letter 0]

25 Apr 2024

Response to Reviewers

Title of the Manuscript: Analysis of Contemporary Value and Influence of Intangible Cultural Heritage Based on Online Review Mining

Manuscript ID: PONE-D-24-01376

Dear Dr. Hsu,

We appreciate the time and effort that the reviewers have dedicated to critiquing our manuscript. We have carefully considered each comment and have made several revisions to the manuscript accordingly. Below, we provide a detailed response to each of the comments raised by the reviewers.

To academic editor’ comments:

Comment 1:

Response:

 Thank you for your notice. We have updated the manuscript to meet the requirements of PLOS ONE’s style.

Comment 2:

Please note that PLOS ONE has specific guidelines on code sharing for submissions in which author-generated code underpins the findings in the manuscript. In these cases, all author-generated code must be made available without restrictions upon publication of the work.

Response:

 Thank you for your advice. We have uploaded and shared the code and it can be accessed by DOI: 10.5281/zenodo.11004055

Comment 3:

Response:

 Thank you for your notice. We have updated the founder’s role and contribution in the cover letter. In case it was not seen, we also state it separately here.

1.the National Natural Science Foundation of China (No. 72104219): Study design, data collection and analysis

2.the National Natural Science Foundation of China (No. 62207016): data collection and analysis

3.the MOE Project of Humanities and Social Sciences (No. 21YJC870013): preparation of the manuscript

4.Major Humanities and Social Sciences Research Projects in Zhejiang Higher Education Institutions (No. 2023QN129): preparation of the manuscript, decision to publish.

Comment 4:

In the online submission form, you indicated that your data will be submitted to a repository upon acceptance. We strongly recommend all authors deposit their data before acceptance, as the process can be lengthy and hold up publication timelines. Please note that, though access restrictions are acceptable now, your entire minimal dataset will need to be made freely accessible if your manuscript is accepted for publication. This policy applies to all data except where public deposition would breach compliance with the protocol approved by your research ethics board. If you are unable to adhere to our open data policy, please kindly revise your statement to explain your reasoning and we will seek the editor's input on an exemption. 

Response:

 Thank you for your advice. We have uploaded and shared the data and it can be accessed by DOI: 10.5281/zenodo.11004089.

Comment 5:

We note that Figures 4 and 5 in your submission contain copyrighted images. All PLOS content is published under the Creative Commons Attribution License (CC BY 4.0), which means that the manuscript, images, and Supporting Information files will be freely available online, and any third party is permitted to access, download, copy, distribute, and use these materials in any way, even commercially, with proper attribution. 

Response:

 Thank you for your notice. This two figures Figure 4 and 5 have been generated and edited through a public data visualization platform: https://wordart.com/. The data used are all our own data and have been set to public access. So, there is no copyright issue to be concerned. 

To reviewers' comments:

Reviewer #1: 

Comment:

Accept the Article entitled "Analysis of Contemporary Value and Influence of Intangible Cultural Heritage Based on Online Review Mining". This article addressing Influence of Intangible Cultural Heritage Based on Online Review Mining.

Response:

Thank you for your recognition of our research.

Reviewer #2: 

Comment:

Valuable paper for this era. The examination of ICH's current significance and impact reveals its varied function in fostering innovation, social cohesiveness, cultural diversity, and sustainable development. Communities may take advantage of ICH's potential to improve lives and build a more inclusive and sustainable future by realising its significance and supporting its promotion and preservation.

Response:

Thank you for your recognition of our research.

Reviewer #3: 

Comment:

The authors have used machine learning techniques for ICH (Intangible Cultural Heritage) value perception classification task. The manuscript lacks substantial data and methodology, such as details on neural network architectures (e.g., BERT, Bi-GRU RNN).

Response:

Thank you for your precious advice. We have modified and improved the BERT-based word vector technology and the neural network structure involved in the classification model, especially the core structure of the GRU unit.

We believe the revisions we have made address the reviewers’ concerns comprehensively and enhance the manuscript significantly. Attached, you will find the revised manuscript along with a marked-up copy highlighting the changes made.

We thank you for the opportunity to revise our manuscript and hope that the changes meet your and the reviewers' satisfaction.

Best regards,

Dr. Chao Ma

College of Economic and Management, Zhejiang Normal University

Machao456@hotmail.com

---

## [Decision Letter · Decision Letter 1]

6 Aug 2024

PONE-D-24-01376R1Analysis of Contemporary Value and Influence of Intangible Cultural Heritage Based on Online Review Mining

PLOS ONE

Dear Dr. Ma,

Thank you for submitting your manuscript to PLOS ONE. After careful consideration, we feel that it has merit but does not fully meet PLOS ONE’s publication criteria as it currently stands. Therefore, we invite you to submit a revised version of the manuscript that addresses the points raised during the review process.

We look forward to receiving your revised manuscript.

Kind regards,

Riccardo Ortale

Academic Editor

PLOS ONE

Additional Editor Comments:

The comments from the Reviewers collectively highlight several critical areas that require improvement in the paper.

These areas include the use of state-of-the-art comparative methods, the improvement of data presentation,

formatting and citations, language quality, and the discussion and impact of the findings.

I invite you to undertake a major revision of the paper, ensuring that all comments from the Reviewers are thoroughly addressed in each of these areas.

Reviewers' comments:

Reviewer's Responses to Questions

**Comments to the Author**

1. If the authors have adequately addressed your comments raised in a previous round of review and you feel that this manuscript is now acceptable for publication, you may indicate that here to bypass the “Comments to the Author” section, enter your conflict of interest statement in the “Confidential to Editor” section, and submit your "Accept" recommendation.

Reviewer #4: All comments have been addressed

Reviewer #5: All comments have been addressed

Reviewer #6: All comments have been addressed

2. Is the manuscript technically sound, and do the data support the conclusions?

Reviewer #4: Yes

Reviewer #5: Yes

Reviewer #6: Yes

3. Has the statistical analysis been performed appropriately and rigorously? 

Reviewer #4: No

Reviewer #5: Yes

Reviewer #6: Yes

4. Have the authors made all data underlying the findings in their manuscript fully available?

Reviewer #4: No

Reviewer #5: Yes

Reviewer #6: Yes

5. Is the manuscript presented in an intelligible fashion and written in standard English?

Reviewer #4: Yes

Reviewer #5: Yes

Reviewer #6: Yes

6. Review Comments to the Author

Reviewer #4: 1. The comparative methods are not SOTA. Please add some SOTA classification methods to compare.

2. Four figures of word cloud may not depict your results. add some Tables to show your high-frequency words.

Reviewer #5: The proposed BiGRU-Attention model, enhanced with BERT word embeddings, aims to achieve effective text contemporary value perception recognition based on user-generated content.

All comments have been answered.

Reviewer #6: This study analyzed real online reviews from the Bilibili website. The design model classified the contemporary values of intangible cultural heritage.The study is instructive, but requires minor revisions.

1. Ensure the quality of article formatting and reference citation.

2. A thorough review and revision for clarity, grammar, and overall language quality are necessary.

3. The discussion of the research results is insufficient. Explain how the results contribute to the existing body of knowledge and highlight their significance.

4. Describe how the findings can be applied in real-world scenarios and their potential impact on the field.

7. PLOS authors have the option to publish the peer review history of their article (what does this mean?). If published, this will include your full peer review and any attached files.

Reviewer #4: No

Reviewer #5: No

Reviewer #6: No

---

## [Author Response · Author response to Decision Letter 1]

10 Sep 2024

Response to Reviewers

Title of the Manuscript: Analysis of Contemporary Value and Influence of Intangible Cultural Heritage Based on Online Review Mining

Manuscript ID: PONE-D-24-01376R1

Dear Dr. Riccardo Ortale,

We appreciate the time and effort that the reviewers have dedicated to critiquing our manuscript. We have carefully considered each comment and have made several revisions to the manuscript accordingly. Below, we provide a detailed response to each of the comments raised by the reviewers.

To reviewers' comments:

Reviewer #4: 

Comment:

1. The comparative methods are not SOTA. Please add some SOTA classification methods to compare.

2. Four figures of word cloud may not depict your results. add some Tables to show your high-frequency words.

Response:

Thank you for your precious advice.

1. According to your first comment, we added two classification methods experiment, the BERT pre-train model and the TEXT-RCNN model are selected as the SOTA methods for comparison.

2. According to your second comment, we added tables to present high-frequency keywords from each category.

Thank you again for your careful review of our manuscript, and we hope you will be satisfied with the revision.

Reviewer #5: 

Comment:

The proposed BiGRU-Attention model, enhanced with BERT word embeddings, aims to achieve effective text contemporary value perception recognition based on user-generated content.

All comments have been answered.

Response:

Thank you for your recognition of our research.

Reviewer #6: 

Comment:

This study analyzed real online reviews from the Bilibili website. The design model classified the contemporary values of intangible cultural heritage. The study is instructive, but requires minor revisions.

1. Ensure the quality of article formatting and reference citation.

2. A thorough review and revision for clarity, grammar, and overall language quality are necessary.

3. The discussion of the research results is insufficient. Explain how the results contribute to the existing body of knowledge and highlight their significance.

4. Describe how the findings can be applied in real-world scenarios and their potential impact on the field.

Response:

Thank you for your precious advice.

1. According to your first and second comments, we have reviewed and revised our text and format. We have carefully checked the format requirements of the journal and improved them. We hope that you will be satisfied.

2. According to your third and fourth comments, we have added some discussion about the implications of this study for the existing knowledge system and the role it can play in practical environments in the conclusion of the article. The additional discussion has been marked in red, and we hope you will be satisfied.

Thank you again for your careful review of our manuscript.

We believe the revisions we have made address the reviewers’ concerns comprehensively and enhance the manuscript significantly. Attached, you will find the revised manuscript along with a marked-up copy highlighting the changes made.

We thank you for the opportunity to revise our manuscript and hope that the changes meet your and the reviewers' satisfaction.

Best regards,

Dr. Chao Ma

College of Economic and Management, Zhejiang Normal University

Machao456@hotmail.com

---

## [Decision Letter · Decision Letter 2]

29 Sep 2024

PONE-D-24-01376R2Analysis of Contemporary Value and Influence of Intangible Cultural Heritage Based on Online Review MiningPLOS ONE

Dear Dr. Ma,

Thank you for submitting your manuscript to PLOS ONE. After careful consideration, we feel that it has merit but does not fully meet PLOS ONE’s publication criteria as it currently stands. Therefore, we invite you to submit a revised version of the manuscript that addresses the points raised during the review process.

We look forward to receiving your revised manuscript.

Kind regards,

Riccardo Ortale

Academic Editor

PLOS ONE

Journal Requirements:

Additional Editor Comments:

Based on the feedback received, I invite you to resubmit a minor revision of your manuscript that carefully addresses all Reviewer comments.

Reviewers' comments:

Reviewer's Responses to Questions

**Comments to the Author**

1. If the authors have adequately addressed your comments raised in a previous round of review and you feel that this manuscript is now acceptable for publication, you may indicate that here to bypass the “Comments to the Author” section, enter your conflict of interest statement in the “Confidential to Editor” section, and submit your "Accept" recommendation.

Reviewer #4: All comments have been addressed

Reviewer #6: All comments have been addressed

2. Is the manuscript technically sound, and do the data support the conclusions?

Reviewer #4: (No Response)

Reviewer #6: Partly

3. Has the statistical analysis been performed appropriately and rigorously? 

Reviewer #4: (No Response)

Reviewer #6: Yes

4. Have the authors made all data underlying the findings in their manuscript fully available?

Reviewer #4: Yes

Reviewer #6: Yes

5. Is the manuscript presented in an intelligible fashion and written in standard English?

Reviewer #4: Yes

Reviewer #6: Yes

6. Review Comments to the Author

Reviewer #4: (No Response)

Reviewer #6: The authors have basically solved the existing problems and has some suggestions. Why did the authors mark cultural value, social value, and economic value? What is the basis of this classification?

7. PLOS authors have the option to publish the peer review history of their article (what does this mean?). If published, this will include your full peer review and any attached files.

Reviewer #4: No

Reviewer #6: No

---

## [Author Response · Author response to Decision Letter 2]

10 Oct 2024

Response to Reviewers

Title of the Manuscript: Analysis of Contemporary Value and Influence of Intangible Cultural Heritage Based on Online Review Mining

Manuscript ID: PONE-D-24-01376R2

Dear Dr. Riccardo Ortale,

We appreciate the time and effort that the reviewers have dedicated to critiquing our manuscript. We have carefully considered each comment and have made several revisions to the manuscript accordingly. Below, we provide a detailed response to each of the comments raised by the reviewers.

To reviewers' comments:

Reviewer #6: 

Comment:

The authors have basically solved the existing problems and has some suggestions. Why did the authors mark cultural value, social value, and economic value? What is the basis of this classification?

Response:

Thank you for your careful review. Based on your valuable suggestions, we have added a discussion on the source of classification, and we have added a theoretical basis of Hofstadter's cultural value and related references. This is the basis for the classification of the perception of intangible cultural heritage value.

We believe the revisions we have made address the reviewers’ concerns comprehensively and enhance the manuscript significantly. Attached, you will find the revised manuscript along with a marked-up copy highlighting the changes made.

We thank you for the opportunity to revise our manuscript and hope that the changes meet your and the reviewers' satisfaction.

Best regards,

Dr. Chao Ma

College of Economic and Management, Zhejiang Normal University

Machao456@hotmail.com

---

## [Decision Letter · Decision Letter 3]

22 Nov 2024

PONE-D-24-01376R3Analysis of Contemporary Value and Influence of Intangible Cultural Heritage Based on Online Review MiningPLOS ONE

Dear Dr. Ma,

Thank you for submitting your manuscript to PLOS ONE. After careful consideration, we feel that it has merit but does not fully meet PLOS ONE’s publication criteria as it currently stands. Therefore, we invite you to submit a revised version of the manuscript that addresses the points raised during the review process.

We look forward to receiving your revised manuscript.

Kind regards,

Riccardo Ortale

Academic Editor

PLOS ONE

Journal Requirements:

**Additional Editor Comments:**

Based on the feedback received, I invite you to prepare and submit a minor revision of your paper that carefully addresses all comments from the Reviewers in detail.

Regarding the suggested previously published works, while it is advisable to review them and assess their relevance to your research, please note that citing them is not mandatory.

Reviewers' comments:

Reviewer's Responses to Questions

**Comments to the Author**

1. If the authors have adequately addressed your comments raised in a previous round of review and you feel that this manuscript is now acceptable for publication, you may indicate that here to bypass the “Comments to the Author” section, enter your conflict of interest statement in the “Confidential to Editor” section, and submit your "Accept" recommendation.

Reviewer #4: All comments have been addressed

Reviewer #7: (No Response)

2. Is the manuscript technically sound, and do the data support the conclusions?

Reviewer #4: Yes

Reviewer #7: Yes

3. Has the statistical analysis been performed appropriately and rigorously? 

Reviewer #4: Yes

Reviewer #7: Yes

4. Have the authors made all data underlying the findings in their manuscript fully available?

Reviewer #4: Yes

Reviewer #7: No

5. Is the manuscript presented in an intelligible fashion and written in standard English?

Reviewer #4: Yes

Reviewer #7: Yes

6. Review Comments to the Author

Reviewer #4: (No Response)

Reviewer #7: General Overview: The manuscript presents an innovative approach to analyzing how intangible cultural heritage (ICH) is perceived and valued through the analysis of online reviews. It employs data mining methods and deep learning models, such as BiGRU with attention, to identify the cultural, social, and economic dimensions of ICH value.

Strengths:

Advanced Methodology: The use of the BiGRU-Attention model with BERT-based embeddings is relevant and well-justified, providing an in-depth semantic analysis.

Multidimensional Approach: The segmentation of value perceptions into cultural, social, and economic adds depth to the analysis and allows for a comprehensive understanding of ICH in the contemporary context.

Areas for Improvement:

Expansion of the Theoretical Framework: While the manuscript provides a solid theoretical context, including additional studies on the interaction of ICH with legal, economic, and cultural frameworks would strengthen the argument.

Citation Recommendations:

Heredia-Carroza, J., Palma Martos, L. A., & Aguado, L. F. (2023). Does copyright understand intangible heritage? The case of flamenco in Spain. International Journal of Heritage Studies, 29(6). This article addresses the challenge of protecting ICH through copyright and examines how these regulations impact the perception and value of heritage. Its inclusion would provide valuable legal context for the analysis of ICH in digital platforms.

Heredia-Carroza, J., Palma, L., de Sancha-Navarro, J., & Chavarría-Ortiz, C. (2023). Consumption Habits of Recorded Music: Determinants of Flamenco Albums Acquisition. SAGE OPEN, 13(3). This study analyses the consumption habits of traditional music, offering relevant insights into the economic factors influencing the acquisition and perceived value of ICH.

Heredia-Carroza, J., Saraiva, H., & Chavarría-Ortiz, C. (2021). How to measure flamenco performer value? A cultural economic approach. Scientific Annals of Economics and Business, 68/SI, 71-77. This paper presents an economic approach to measuring the value of flamenco performers, providing a quantitative perspective that could enrich the discussion on the economic impacts of ICH.

Han, J. H., & Bae, S. Y. (2022). Roles of authenticity and nostalgia in cultural heritage tourists’ travel experience sharing behavior on social media. Asia Pacific Journal of Tourism Research, 27(4), 411–427. This article offers an interesting perspective on how authenticity and nostalgia influence tourists' behavior in sharing ICH experiences on social media. Including it could enhance the discussion on how emotions and perceptions affect the value and dissemination of ICH.

Deeper Discussion: The discussion section could benefit from more comparisons with previous studies. Integrating the recommended references would help better position the findings within the current research context and demonstrate how online review analysis contributes to understanding ICH from an interdisciplinary approach.

Practical Implications and Recommendations: Including a section that discusses the practical implications of the findings for cultural policymakers, heritage managers, and academics would be valuable. This section could highlight how the results could inform the preservation of ICH and its integration into the digital environment.

Limitations and Future Research: Although the manuscript briefly touches on limitations, expanding this section to include specific aspects related to potential biases in online reviews and how they might influence the representation of ICH would be beneficial. Future research directions could explore the application of other data analysis techniques to validate and extend the results.

Conclusion: The manuscript is a valuable contribution to the study of contemporary perceptions of ICH through digital data analysis. Citing the recommended studies would strengthen the theoretical framework and enrich the discussion, aligning the work with current debates on the legal, cultural, and economic aspects of intangible heritage.

7. PLOS authors have the option to publish the peer review history of their article (what does this mean?). If published, this will include your full peer review and any attached files.

Reviewer #4: No

Reviewer #7: No

---

## [Author Response · Author response to Decision Letter 3]

27 Nov 2024

Response to Reviewers

Title of the Manuscript: Analysis of Contemporary Value and Influence of Intangible Cultural Heritage Based on Online Review Mining

Manuscript ID: PONE-D-24-01376R3

Dear Dr. Riccardo Ortale,

We appreciate the time and effort that the reviewers have dedicated to critiquing our manuscript. We have carefully considered each comment and have made several revisions to the manuscript accordingly. Below, we provide a detailed response to each of the comments raised by the reviewers.

To reviewers' comments:

Reviewer #7: 

General Overview

Comment: The manuscript presents an innovative approach to analyzing how intangible cultural heritage (ICH) is perceived and valued through the analysis of online reviews. It employs data mining methods and deep learning models, such as BiGRU with attention, to identify the cultural, social, and economic dimensions of ICH value.

Response: Thank you for your positive feedback on the methodology and the innovative approach of the study. We are pleased that you found the analysis of online reviews and the use of advanced models like BiGRU with attention appropriate for identifying the cultural, social, and economic dimensions of ICH value.

Strengths：

Advanced Methodology

Comment: The use of the BiGRU-Attention model with BERT-based embeddings is relevant and well-justified, providing an in-depth semantic analysis.

Response: We are grateful for your acknowledgment of the BiGRU-Attention model and the BERT-based embeddings. We believe that these advanced methods significantly enhance semantic analysis, and we have made minor clarifications in the revised manuscript to ensure the methodology’s application is fully explained.

Multidimensional Approach

Comment: The segmentation of value perceptions into cultural, social, and economic adds depth to the analysis and allows for a comprehensive understanding of ICH in the contemporary context.

Response: Thank you for your positive comments on the multidimensional approach. We have added more detail to the explanation of how we define and categorize cultural, social, and economic dimensions in the revised manuscript to enhance clarity and understanding.

Areas for Improvement：

Expansion of the Theoretical Framework

Comment: While the manuscript provides a solid theoretical context, including additional studies on the interaction of ICH with legal, economic, and cultural frameworks would strengthen the argument.

Response: We appreciate your suggestion to expand the theoretical framework. In response, we have included several additional studies that explore the interaction of ICH with legal, economic, and cultural frameworks. These additions further strengthen the theoretical context and provide a more comprehensive view of ICH in the digital age.

Citation Recommendations

Comment:

Heredia-Carroza, J., Palma Martos, L. A., & Aguado, L. F. (2023). This article addresses the challenge of protecting ICH through copyright and examines how these regulations impact the perception and value of heritage.

Heredia-Carroza, J., Palma, L., de Sancha-Navarro, J., & Chavarría-Ortiz, C. (2023). This study analyzes the consumption habits of traditional music, offering relevant insights into the economic factors influencing the acquisition and perceived value of ICH.

Heredia-Carroza, J., Saraiva, H., & Chavarría-Ortiz, C. (2021). This paper presents an economic approach to measuring the value of flamenco performers, providing a quantitative perspective that could enrich the discussion on the economic impacts of ICH.

Han, J. H., & Bae, S. Y. (2022). This article offers an interesting perspective on how authenticity and nostalgia influence tourists' behavior in sharing ICH experiences on social media.

Response: We greatly appreciate the citation recommendations you provided. We have carefully reviewed these articles and integrated them into the revised manuscript. These references have been incorporated in relevant sections to provide a more interdisciplinary perspective, as you suggested.

Deeper Discussion

Comment: The discussion section could benefit from more comparisons with previous studies. Integrating the recommended references would help better position the findings within the current research context and demonstrate how online review analysis contributes to understanding ICH from an interdisciplinary approach.

Response: Thank you for this insightful suggestion. We have expanded the discussion section to include comparisons with relevant studies, particularly focusing on how our findings align with or differ from existing research in ICH, cultural economics, and digital heritage studies. The newly integrated references enrich our discussion and situate our work within a broader scholarly context, particularly in relation to legal, economic, and cultural perspectives on ICH.

Practical Implications and Recommendations

Comment: Including a section that discusses the practical implications of the findings for cultural policymakers, heritage managers, and academics would be valuable. This section could highlight how the results could inform the preservation of ICH and its integration into the digital environment.

Response: We agree that discussing the practical implications of our findings is essential. We have added a new section on Practical Implications and Recommendations, where we highlight how our study can inform cultural policymakers, heritage managers, and academics. We discuss how digital platforms can be leveraged to preserve and promote ICH, the importance of balancing copyright with public access, and how the findings can guide heritage management strategies in the digital age. We believe this addition significantly enhances the manuscript’s relevance for practitioners.

Limitations and Future Research

Comment: Although the manuscript briefly touches on limitations, expanding this section to include specific aspects related to potential biases in online reviews and how they might influence the representation of ICH would be beneficial. Future research directions could explore the application of other data analysis techniques to validate and extend the results.

Response: Thank you for your valuable feedback on the limitations section. We have expanded this section to discuss potential biases in online reviews, such as demographic factors, emotional biases, and platform-specific algorithms, which may influence the representation of ICH. Additionally, we have suggested future research directions, including the application of advanced techniques such as sentiment analysis and machine learning to validate our findings and address the limitations of our current approach. These additions provide a clearer direction for future work in this area.

Conclusion

Comment: The manuscript is a valuable contribution to the study of contemporary perceptions of ICH through digital data analysis. Citing the recommended studies would strengthen the theoretical framework and enrich the discussion, aligning the work with current debates on the legal, cultural, and economic aspects of intangible heritage.

Response: We sincerely appreciate your positive assessment of our manuscript. As mentioned, we have incorporated the recommended references to strengthen the theoretical framework and deepen the interdisciplinary discussion on ICH. These additions have helped position our work more clearly within the current academic debates on ICH's legal, economic, and cultural dimensions.

We believe the revisions we have made address the reviewers’ concerns comprehensively and enhance the manuscript significantly. Attached, you will find the revised manuscript along with a marked-up copy highlighting the changes made.

We thank you for the opportunity to revise our manuscript and hope that the changes meet your and the reviewers' satisfaction.

Best regards,

Dr. Chao Ma

College of Economic and Management, Zhejiang Normal University

machao456@hotmail.com

---

## [Decision Letter · Decision Letter 4]

3 Dec 2024

Analysis of Contemporary Value and Influence of Intangible Cultural Heritage Based on Online Review Mining

PONE-D-24-01376R4

Dear Dr. Ma,

We’re pleased to inform you that your manuscript has been judged scientifically suitable for publication and will be formally accepted for publication once it meets all outstanding technical requirements.

Kind regards,

Riccardo Ortale

Academic Editor

PLOS ONE

Additional Editor Comments (optional):

Reviewers' comments:

Reviewer's Responses to Questions

**Comments to the Author**

1. If the authors have adequately addressed your comments raised in a previous round of review and you feel that this manuscript is now acceptable for publication, you may indicate that here to bypass the “Comments to the Author” section, enter your conflict of interest statement in the “Confidential to Editor” section, and submit your "Accept" recommendation.

Reviewer #4: All comments have been addressed

Reviewer #7: All comments have been addressed

2. Is the manuscript technically sound, and do the data support the conclusions?

Reviewer #4: Yes

Reviewer #7: Yes

3. Has the statistical analysis been performed appropriately and rigorously? 

Reviewer #4: Yes

Reviewer #7: Yes

4. Have the authors made all data underlying the findings in their manuscript fully available?

Reviewer #4: Yes

Reviewer #7: Yes

5. Is the manuscript presented in an intelligible fashion and written in standard English?

Reviewer #4: Yes

Reviewer #7: Yes

6. Review Comments to the Author

Reviewer #4: All concerns about dual publication, research ethics, or publication ethics have been addressed. I suggest to publish this paper in PLOS One!

Reviewer #7: My comments were well addressed by the author. I recommend the publication of the paper. Congratulations on this excellent work.

7. PLOS authors have the option to publish the peer review history of their article (what does this mean?). If published, this will include your full peer review and any attached files.

Reviewer #4: No

Reviewer #7: **Yes: **Jesús Heredia-Carroza

---

## [Editor Report · Acceptance letter]

10 Dec 2024

PONE-D-24-01376R4 

PLOS ONE

Dear Dr. Ma, 

I'm pleased to inform you that your manuscript has been deemed suitable for publication in PLOS ONE. Congratulations! Your manuscript is now being handed over to our production team.

Kind regards, 

on behalf of

Dr. Riccardo Ortale 

Academic Editor

PLOS ONE